# Streaming Weak Submodularity:
# Interpreting Neural Networks on the Fly

**Ethan R. Elenberg**
Department of Electrical
and Computer Engineering
The University of Texas at Austin
elenberg@utexas.edu

**Alexandros G. Dimakis**
Department of Electrical
and Computer Engineering
The University of Texas at Austin
dimakis@austin.utexas.edu

**Moran Feldman**
Department of Mathematics
and Computer Science
Open University of Israel
moranfe@openu.ac.il

**Amin Karbasi**
Department of Electrical Engineering
Department of Computer Science
Yale University
amin.karbasi@yale.edu

## Abstract

In many machine learning applications, it is important to explain the predictions of a black-box classifier. For example, why does a deep neural network assign an image to a particular class? We cast interpretability of black-box classifiers as a combinatorial maximization problem and propose an efficient streaming algorithm to solve it subject to cardinality constraints. By extending ideas from Badanidiyuru et al. [2014], we provide a constant factor approximation guarantee for our algorithm in the case of random stream order and a *weakly submodular* objective function. This is the first such theoretical guarantee for this general class of functions, and we also show that no such algorithm exists for a worst case stream order. Our algorithm obtains similar explanations of Inception V3 predictions 10 times faster than the state-of-the-art LIME framework of Ribeiro et al. [2016].

## 1 Introduction

Consider the following combinatorial optimization problem. Given a ground set $\mathcal{N}$ of $N$ elements and a set function $f \colon 2^{\mathcal{N}} \mapsto \mathbb{R}_{\geq 0}$, find the set $S$ of size $k$ which maximizes $f(S)$. This formulation is at the heart of many machine learning applications such as sparse regression, data summarization, facility location, and graphical model inference. Although the problem is intractable in general, if $f$ is assumed to be *submodular* then many approximation algorithms have been shown to perform provably within a constant factor from the best solution.

Some disadvantages of the standard greedy algorithm of Nemhauser et al. [1978] for this problem are that it requires repeated access to each data element and a large total number of function evaluations. This is undesirable in many large-scale machine learning tasks where the entire dataset cannot fit in main memory, or when a single function evaluation is time consuming. In our main application, each function evaluation corresponds to inference on a large neural network and can take a few seconds. In contrast, streaming algorithms make a small number of passes (often only one) over the data and have sublinear space complexity, and thus, are ideal for tasks of the above kind.

Recent ideas, algorithms, and techniques from submodular set function theory have been used to derive similar results in much more general settings. For example, Elenberg et al. [2016a] used the concept of *weak submodularity* to derive approximation and parameter recovery guarantees for

nonlinear sparse regression. Thus, a natural question is whether recent results on streaming algorithms for maximizing submodular functions [Badanidiyuru et al., 2014, Buchbinder et al., 2015, Chekuri et al., 2015] extend to the weakly submodular setting.

This paper answers the above question by providing the first analysis of a streaming algorithm for any class of approximately submodular functions. We use key algorithmic components of SIEVE-STREAMING [Badanidiyuru et al., 2014], namely greedy thresholding and binary search, combined with a novel analysis to prove a constant factor approximation for $\gamma$-weakly submodular functions (defined in Section 3). Specifically, our contributions are as follows.

- An impossibility result showing that, even for 0.5-weakly submodular objectives, no randomized streaming algorithm which uses $o(N)$ memory can have a constant approximation ratio when the ground set elements arrive in a worst case order.
- STREAK: a greedy, deterministic streaming algorithm for maximizing $\gamma$-weakly submodular functions which uses $\mathcal{O}(\varepsilon^{-1}k\log k)$ memory and has an approximation ratio of $(1 - \varepsilon)\frac{\gamma}{2} \cdot (3 - e^{-\gamma/2} - 2\sqrt{2 - e^{-\gamma/2}})$ when the ground set elements arrive in a random order.
- An experimental evaluation of our algorithm in two applications: nonlinear sparse regression using pairwise products of features and interpretability of black-box neural network classifiers.

The above theoretical impossibility result is quite surprising since it stands in sharp contrast to known streaming algorithms for submodular objectives achieving a constant approximation ratio even for worst case stream order.

One advantage of our approach is that, while our approximation guarantees are in terms of $\gamma$, our algorithm STREAK runs without requiring prior knowledge about the value of $\gamma$. This is important since the weak submodularity parameter $\gamma$ is hard to compute, especially in streaming applications, as a single element can alter $\gamma$ drastically.

We use our streaming algorithm for neural network interpretability on Inception V3 [Szegedy et al., 2016]. For that purpose, we define a new set function maximization problem similar to LIME [Ribeiro et al., 2016] and apply our framework to approximately maximize this function. Experimentally, we find that our interpretability method produces explanations of similar quality as LIME, but runs approximately 10 times faster.

## 2   Related Work

Monotone submodular set function maximization has been well studied, starting with the classical analysis of greedy forward selection subject to a matroid constraint [Nemhauser et al., 1978, Fisher et al., 1978]. For the special case of a uniform matroid constraint, the greedy algorithm achieves an approximation ratio of $1 - 1/e$ [Fisher et al., 1978], and a more involved algorithm obtains this ratio also for general matroid constraints [Călinescu et al., 2011]. In general, no polynomial-time algorithm can have a better approximation ratio even for a uniform matroid constraint [Nemhauser and Wolsey, 1978, Feige, 1998]. However, it is possible to improve upon this bound when the data obeys some additional guarantees [Conforti and Cornuéjols, 1984, Vondrák, 2010, Sviridenko et al., 2015]. For maximizing nonnegative, not necessarily monotone, submodular functions subject to a general matroid constraint, the state-of-the-art randomized algorithm achieves an approximation ratio of 0.385 [Buchbinder and Feldman, 2016b]. Moreover, for uniform matroids there is also a deterministic algorithm achieving a slightly worse approximation ratio of $1/e$ [Buchbinder and Feldman, 2016a]. The reader is referred to Bach [2013] and Krause and Golovin [2014] for surveys on submodular function theory.

A recent line of work aims to develop new algorithms for optimizing submodular functions suitable for large-scale machine learning applications. Algorithmic advances of this kind include STOCHASTIC-GREEDY [Mirzasoleiman et al., 2015], SIEVE-STREAMING [Badanidiyuru et al., 2014], and several distributed approaches [Mirzasoleiman et al., 2013, Barbosa et al., 2015, 2016, Pan et al., 2014, Khanna et al., 2017b]. Our algorithm extends ideas found in SIEVE-STREAMING and uses a different analysis to handle more general functions. Additionally, submodular set functions have been used to prove guarantees for online and active learning problems [Hoi et al., 2006, Wei et al., 2015, Buchbinder et al., 2015]. Specifically, in the online setting corresponding to our setting

(*i.e.*, maximizing a monotone function subject to a cardinality constraint), Chan et al. [2017] achieve a competitive ratio of about $0.3178$ when the function is submodular.

The concept of weak submodularity was introduced in Krause and Cevher [2010], Das and Kempe [2011], where it was applied to the specific problem of feature selection in linear regression. Their main results state that if the data covariance matrix is not too correlated (using either incoherence or restricted eigenvalue assumptions), then maximizing the goodness of fit $f(S) = R_S^2$ as a function of the feature set $S$ is weakly submodular. This leads to constant factor approximation guarantees for several greedy algorithms. Weak submodularity was connected with Restricted Strong Convexity in Elenberg et al. [2016a,b]. This showed that the same assumptions which imply the success of regularization also lead to guarantees on greedy algorithms. This framework was later used for additional algorithms and applications [Khanna et al., 2017a,b]. Other approximate versions of submodularity were used for greedy selection problems in Horel and Singer [2016], Hassidim and Singer [2017], Altschuler et al. [2016], Bian et al. [2017]. To the best of our knowledge, this is the first analysis of streaming algorithms for approximately submodular set functions.

Increased interest in interpretable machine learning models has led to extensive study of sparse feature selection methods. For example, Bahmani et al. [2013] consider greedy algorithms for logistic regression, and Yang et al. [2016] solve a more general problem using $\ell_1$ regularization. Recently, Ribeiro et al. [2016] developed a framework called LIME for interpreting black-box neural networks, and Sundararajan et al. [2017] proposed a method that requires access to the network's gradients with respect to its inputs. We compare our algorithm to variations of LIME in Section 6.2.

## 3   Preliminaries

First we establish some definitions and notation. Sets are denoted with capital letters, and all big O notation is assumed to be scaling with respect to $N$ (the number of elements in the input stream). Given a set function $f$, we often use the discrete derivative $f(B \mid A) \triangleq f(A \cup B) - f(A)$. $f$ is monotone if $f(B \mid A) \geq 0, \forall A, B$ and nonnegative if $f(A) \geq 0, \forall A$. Using this notation one can define weakly submodular functions based on the following ratio.

**Definition 3.1** (Weak Submodularity, adapted from Das and Kempe [2011]). *A monotone nonnegative set function* $f : 2^{\mathcal{N}} \mapsto \mathbb{R}_{\geq 0}$ *is called* $\gamma$-*weakly submodular for an integer* $r$ *if*

$$\gamma \leq \gamma_r \triangleq \min_{\substack{L, S \subseteq \mathcal{N}: \\ |L|, |S \setminus L| \leq r}} \frac{\sum_{j \in S \setminus L} f(j \mid L)}{f(S \mid L)} \ ,$$

*where the ratio is considered to be equal to* $1$ *when its numerator and denominator are both* $0$.

This generalizes submodular functions by relaxing the *diminishing returns* property of discrete derivatives. It is easy to show that $f$ is submodular if and only if $\gamma_{|\mathcal{N}|} = 1$.

**Definition 3.2** (Approximation Ratio). *A streaming maximization algorithm* ALG *which returns a set* $S$ *has approximation ratio* $R \in [0, 1]$ *if* $\mathbb{E}[f(S)] \geq R \cdot f(OPT)$, *where* $OPT$ *is the optimal solution and the expectation is over the random decisions of the algorithm and the randomness of the input stream order (when it is random).*

Formally our problem is as follows. Assume that elements from a ground set $\mathcal{N}$ arrive in a stream at either random or worst case order. The goal is then to design a one pass streaming algorithm that given oracle access to a nonnegative set function $f : 2^{\mathcal{N}} \mapsto \mathbb{R}_{\geq 0}$ maintains at most $o(N)$ elements in memory and returns a set $S$ of size at most $k$ approximating

$$\max_{|T| \leq k} f(T) \ ,$$

up to an approximation ratio $R(\gamma_k)$. Ideally, this approximation ratio should be as large as possible, and we also want it to be a function of $\gamma_k$ and nothing else. In particular, we want it to be independent of $k$ and $N$.

To simplify notation, we use $\gamma$ in place of $\gamma_k$ in the rest of the paper. Additionally, **proofs for all our theoretical results are deferred to the Supplementary Material.**

# 4   Impossibility Result

To prove our negative result showing that no streaming algorithm for our problem has a constant approximation ratio against a worst case stream order, we first need to construct a weakly submodular set function $f_k$. Later we use it to construct a bad instance for any given streaming algorithm.

Fix some $k \geq 1$, and consider the ground set $\mathcal{N}_k = \{u_i, v_i\}_{i=1}^k$. For ease of notation, let us define for every subset $S \subseteq \mathcal{N}_k$

$$u(S) = |S \cap \{u_i\}_{i=1}^k| \ , \quad v(S) = |S \cap \{v_i\}_{i=1}^k| \ .$$

Now we define the following set function:

$$f_k(S) = \min\{2 \cdot u(S) + 1, 2 \cdot v(S)\} \quad \forall\, S \subseteq \mathcal{N}_k \ .$$

**Lemma 4.1.** $f_k$ *is nonnegative, monotone and* $0.5$*-weakly submodular for the integer* $|\mathcal{N}_k|$.

Since $|\mathcal{N}_k| = 2k$, the maximum value of $f_k$ is $f_k(\mathcal{N}_k) = 2 \cdot v(\mathcal{N}_k) = 2k$. We now extend the ground set of $f_k$ by adding to it an arbitrary large number $d$ of dummy elements which do not affect $f_k$ at all. Clearly, this does not affect the properties of $f_k$ proved in Lemma 4.1. However, the introduction of dummy elements allows us to assume that $k$ is an arbitrary small value compared to $N$, which is necessary for the proof of the next theorem. In a nutshell, this proof is based on the observation that the elements of $\{u_i\}_{i=1}^k$ are indistinguishable from the dummy elements as long as no element of $\{v_i\}_{i=1}^k$ has arrived yet.

**Theorem 4.2.** *For every constant* $c \in (0, 1]$ *there is a large enough* $k$ *such that no randomized streaming algorithm that uses* $o(N)$ *memory to solve* $\max_{|S| \leq 2k} f_k(S)$ *has an approximation ratio of* $c$ *for a worst case stream order.*

We note that $f_k$ has strong properties. In particular, Lemma 4.1 implies that it is $0.5$-weakly submodular for every $0 \leq r \leq |\mathcal{N}|$. In contrast, the algorithm we show later assumes weak submodularity only for the cardinality constraint $k$. Thus, the above theorem implies that worst case stream order precludes a constant approximation ratio even for functions with much stronger properties compared to what is necessary for getting a constant approximation ratio when the order is random.

The proof of Theorem 4.2 relies critically on the fact that each element is seen exactly once. In other words, once the algorithm decides to discard an element from its memory, this element is gone forever, which is a standard assumption for streaming algorithms. Thus, the theorem does not apply to algorithms that use multiple passes over $\mathcal{N}$, or non-streaming algorithms that use $o(N)$ writable memory, and their analysis remains an interesting open problem.

# 5   Streaming Algorithms

In this section we give a deterministic streaming algorithm for our problem which works in a model in which the stream contains the elements of $\mathcal{N}$ in a random order. We first describe in Section 5.1 such a streaming algorithm assuming access to a value $\tau$ which approximates $a\gamma \cdot f(OPT)$, where $a$ is a shorthand for $a = (\sqrt{2 - e^{-\gamma/2}} - 1)/2$. Then, in Section 5.2 we explain how this assumption can be removed to obtain STREAK and bound its approximation ratio, space complexity, and running time.

## 5.1   Algorithm with access to $\tau$

Consider Algorithm 1. In addition to the input instance, this algorithm gets a parameter $\tau \in [0, a\gamma \cdot f(OPT)]$. One should think of $\tau$ as close to $a\gamma \cdot f(OPT)$, although the following analysis of the algorithm does not rely on it. We provide an outline of the proof, but defer the technical details to the Supplementary Material.

**Theorem 5.1.** *The expected value of the set produced by Algorithm 1 is at least*

$$\frac{\tau}{a} \cdot \frac{3 - e^{-\gamma/2} - 2\sqrt{2 - e^{-\gamma/2}}}{2} = \tau \cdot \left(\sqrt{2 - e^{-\gamma/2}} - 1\right) \ .$$

---

**Algorithm 1** THRESHOLD GREEDY$(f, k, \tau)$

---

Let $S \leftarrow \varnothing$.
**while** there are more elements **do**
    Let $u$ be the next element.
    **if** $|S| < k$ and $f(u \mid S) \geq \tau/k$ **then**
        Update $S \leftarrow S \cup \{u\}$.
    **end if**
**end while**
**return:** $S$

---

---

**Algorithm 2** STREAK$(f, k, \varepsilon)$

---

Let $m \leftarrow 0$, and let $I$ be an (originally empty) collection of instances of Algorithm 1.
**while** there are more elements **do**
    Let $u$ be the next element.
    **if** $f(u) \geq m$ **then**
        Update $m \leftarrow f(u)$ and $u_m \leftarrow u$.
    **end if**
    Update $I$ so that it contains an instance of Algorithm 1 with $\tau = x$ for every $x \in \{(1-\varepsilon)^i \mid i \in \mathbb{Z}$ and $(1-\varepsilon)m/(9k^2) \leq (1-\varepsilon)^i \leq mk\}$, as explained in Section 5.2.
    Pass $u$ to all instances of Algorithm 1 in $I$.
**end while**
**return:** the best set among all the outputs of the instances of Algorithm 1 in $I$ and the singleton set $\{u_m\}$.

---

*Proof (Sketch).* Let $\mathcal{E}$ be the event that $f(S) < \tau$, where $S$ is the output produced by Algorithm 1. Clearly $f(S) \geq \tau$ whenever $\mathcal{E}$ does not occur, and thus, it is possible to lower bound the expected value of $f(S)$ using $\mathcal{E}$ as follows.

**Observation 5.2.** *Let $S$ denote the output of Algorithm 1, then $\mathbb{E}[f(S)] \geq (1 - \Pr[\mathcal{E}]) \cdot \tau$.*

The lower bound given by Observation 5.2 is decreasing in $\Pr[\mathcal{E}]$. Proposition 5.4 provides another lower bound for $\mathbb{E}[f(S)]$ which increases with $\Pr[\mathcal{E}]$. An important ingredient of the proof of this proposition is the next observation, which implies that the solution produced by Algorithm 1 is always of size smaller than $k$ when $\mathcal{E}$ happens.

**Observation 5.3.** *If at some point Algorithm 1 has a set $S$ of size $k$, then $f(S) \geq \tau$.*

The proof of Proposition 5.4 is based on the above observation and on the observation that the random arrival order implies that every time that an element of $OPT$ arrives in the stream we may assume it is a random element out of all the $OPT$ elements that did not arrive yet.

**Proposition 5.4.** *For the set $S$ produced by Algorithm 1,*

$$\mathbb{E}[f(S)] \geq \frac{1}{2} \cdot \left( \gamma \cdot [\Pr[\mathcal{E}] - e^{-\gamma/2}] \cdot f(OPT) - 2\tau \right) \ .$$

The theorem now follows by showing that for every possible value of $\Pr[\mathcal{E}]$ the guarantee of the theorem is implied by either Observation 5.2 or Proposition 5.4. Specifically, the former happens when $\Pr[\mathcal{E}] \leq 2 - \sqrt{2 - e^{-\gamma/2}}$ and the later when $\Pr[\mathcal{E}] \geq 2 - \sqrt{2 - e^{-\gamma/2}}$. $\qquad \square$

## 5.2 Algorithm without access to $\tau$

In this section we explain how to get an algorithm which does not depend on $\tau$. Instead, STREAK (Algorithm 2) receives an accuracy parameter $\varepsilon \in (0, 1)$. Then, it uses $\varepsilon$ to run several instances of Algorithm 1 stored in a collection denoted by $I$. The algorithm maintains two variables throughout its execution: $m$ is the maximum value of a singleton set corresponding to an element that the algorithm already observed, and $u_m$ references an arbitrary element satisfying $f(u_m) = m$.

The collection $I$ is updated as follows after each element arrival. If previously $I$ contained an instance of Algorithm 1 with a given value for $\tau$, and it no longer should contain such an instance, then the instance is simply removed. In contrast, if $I$ did not contain an instance of Algorithm 1 with a given value for $\tau$, and it should now contain such an instance, then a new instance with this value for $\tau$ is created. Finally, if $I$ contained an instance of Algorithm 1 with a given value for $\tau$, and it should continue to contain such an instance, then this instance remains in $I$ as is.

**Theorem 5.5.** *The approximation ratio of* STREAK *is at least*

$$(1 - \varepsilon)\gamma \cdot \frac{3 - e^{-\gamma/2} - 2\sqrt{2 - e^{-\gamma/2}}}{2} \ .$$

The proof of Theorem 5.5 shows that in the final collection $I$ there is an instance of Algorithm 1 whose $\tau$ provides a good approximation for $a\gamma \cdot f(OPT)$, and thus, this instance of Algorithm 1 should (up to some technical details) produce a good output set in accordance with Theorem 5.1.

It remains to analyze the space complexity and running time of STREAK. We concentrate on bounding the number of elements STREAK keeps in its memory at any given time, as this amount dominates the space complexity as long as we assume that the space necessary to keep an element is at least as large as the space necessary to keep each one of the numbers used by the algorithm.

**Theorem 5.6.** *The space complexity of* STREAK *is* $\mathcal{O}(\varepsilon^{-1} k \log k)$ *elements.*

The running time of Algorithm 1 is $\mathcal{O}(Nf)$ where, abusing notation, $f$ is the running time of a single oracle evaluation of $f$. Therefore, the running time of STREAK is $\mathcal{O}(Nf\varepsilon^{-1} \log k)$ since it uses at every given time only $\mathcal{O}(\varepsilon^{-1} \log k)$ instances of the former algorithm. Given multiple threads, this can be improved to $\mathcal{O}(Nf + \varepsilon^{-1} \log k)$ by running the $\mathcal{O}(\varepsilon^{-1} \log k)$ instances of Algorithm 1 in parallel.

# 6 Experiments

We evaluate the performance of our streaming algorithm on two sparse feature selection applications.[1] Features are passed to all algorithms in a random order to match the setting of Section 5.

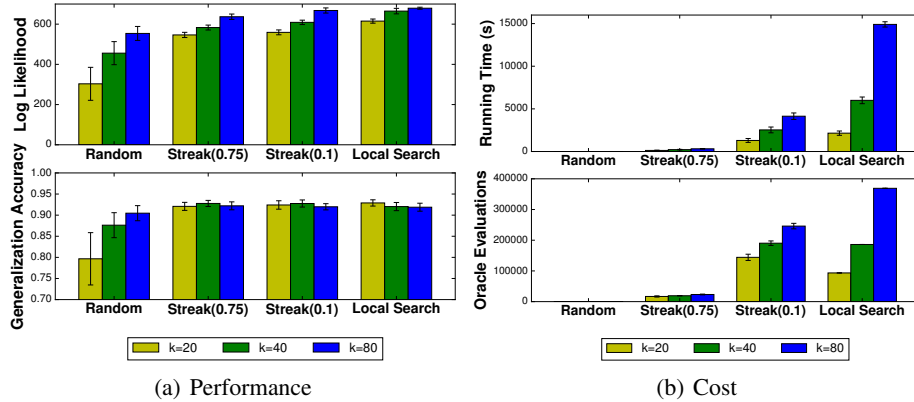

(a) Performance          (b) Cost

Figure 1: Logistic Regression, Phishing dataset with pairwise feature products. Our algorithm is comparable to LOCALSEARCH in both log likelihood and generalization accuracy, with much lower running time and number of model fits in most cases. Results averaged over 40 iterations, error bars show 1 standard deviation.

## 6.1 Sparse Regression with Pairwise Features

In this experiment, a sparse logistic regression is fit on 2000 training and 2000 test observations from the Phishing dataset [Lichman, 2013]. This setup is known to be weakly submodular under mild data assumptions [Elenberg et al., 2016a]. First, the categorical features are one-hot encoded, increasing

[1]Code for these experiments is available at https://github.com/eelenberg/streak.

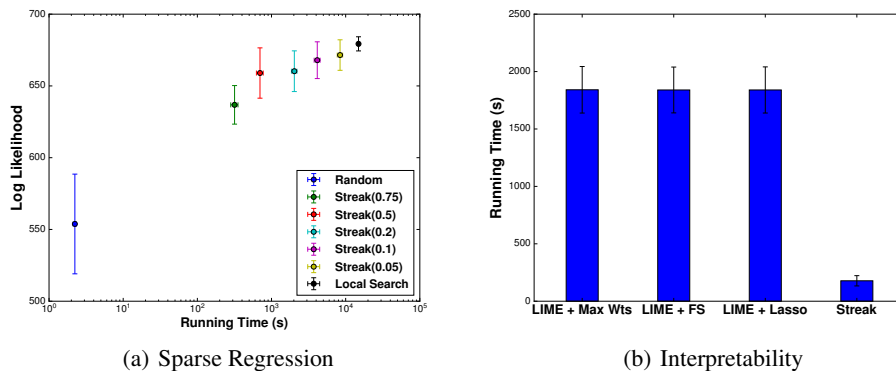

|                          |                          |
|:------------------------:|:------------------------:|
| (a) Sparse Regression    | (b) Interpretability     |

Figure 2: 2(a): Logistic Regression, Phishing dataset with pairwise feature products, $k = 80$ features. By varying the parameter $\varepsilon$, our algorithm captures a time-accuracy tradeoff between RANDOMSUBSET and LOCALSEARCH. Results averaged over 40 iterations, standard deviation shown with error bars. 2(b): Running times of interpretability algorithms on the Inception V3 network, $N = 30$, $k = 5$. Streaming maximization runs 10 times faster than the LIME framework. Results averaged over 40 total iterations using 8 example explanations, error bars show 1 standard deviation.

the feature dimension to 68. Then, all pairwise products are added for a total of $N = 4692$ features. To reduce computational cost, feature products are generated and added to the stream on-the-fly as needed. We compare with 2 other algorithms. RANDOMSUBSET selects the first $k$ features from the random stream. LOCALSEARCH first fills a buffer with the first $k$ features, and then swaps each incoming feature with the feature from the buffer which yields the largest nonnegative improvement.

Figure 1(a) shows both the final log likelihood and the generalization accuracy for RANDOMSUBSET, LOCALSEARCH, and our STREAK algorithm for $\varepsilon = \{0.75, 0.1\}$ and $k = \{20, 40, 80\}$. As expected, the RANDOMSUBSET algorithm has much larger variation since its performance depends highly on the random stream order. It also performs significantly worse than LOCALSEARCH for both metrics, whereas STREAK is comparable for most parameter choices. Figure 1(b) shows two measures of computational cost: running time and the number of oracle evaluations (regression fits). We note STREAK scales better as $k$ increases; for example, STREAK with $k = 80$ and $\varepsilon = 0.1$ ($\varepsilon = 0.75$) runs in about 70% (5%) of the time it takes to run LOCALSEARCH with $k = 40$. Interestingly, our speedups are more substantial with respect to running time. In some cases STREAK actually fits more regressions than LOCALSEARCH, but still manages to be faster. We attribute this to the fact that nearly all of LOCALSEARCH's regressions involve $k$ features, which are slower than many of the small regressions called by STREAK.

Figure 2(a) shows the final log likelihood versus running time for $k = 80$ and $\varepsilon \in [0.05, 0.75]$. By varying the precision $\varepsilon$, we achieve a gradual tradeoff between speed and performance. This shows that STREAK can reduce the running time by over an order of magnitude with minimal impact on the final log likelihood.

## 6.2 Black-Box Interpretability

Our next application is interpreting the predictions of black-box machine learning models. Specifically, we begin with the Inception V3 deep neural network [Szegedy et al., 2016] trained on ImageNet. We use this network for the task of classifying 5 types of flowers via transfer learning. This is done by adding a final softmax layer and retraining the network.

We compare our approach to the LIME framework [Ribeiro et al., 2016] for developing sparse, interpretable explanations. The final step of LIME is to fit a $k$-sparse linear regression in the space of interpretable features. Here, the features are superpixels determined by the SLIC image segmentation algorithm [Achanta et al., 2012] (regions from any other segmentation would also suffice). The number of superpixels is bounded by $N = 30$. After a feature selection step, a final regression is performed on only the selected features. The following feature selection methods are supplied by

LIME: *1. Highest Weights:* fits a full regression and keep the $k$ features with largest coefficients. *2. Forward Selection:* standard greedy forward selection. *3. Lasso:* $\ell_1$ regularization.

We introduce a novel method for black-box interpretability that is similar to but simpler than LIME. As before, we segment an image into $N$ superpixels. Then, for a subset $S$ of those regions we can create a new image that contains only these regions and feed this into the black-box classifier. For a given model $M$, an input image $I$, and a label $\mathbf{L}_1$ we ask for an explanation: why did model $M$ label image $I$ with label $\mathbf{L}_1$. We propose the following solution to this problem. Consider the set function $f(S)$ giving the likelihood that image $I(S)$ has label $\mathbf{L}_1$. We approximately solve

$$\max_{|S| \leq k} f(S) \ ,$$

using STREAK. Intuitively, we are limiting the number of superpixels to $k$ so that the output will include only the most important superpixels, and thus, will represent an interpretable explanation. In our experiments we set $k = 5$.

Note that the set function $f(S)$ depends on the black-box classifier and is neither monotone nor submodular in general. Still, we find that the greedy maximization algorithm produces very good explanations for the flower classifier as shown in Figure 3 and the additional experiments in the Supplementary Material. Figure 2(b) shows that our algorithm is much faster than the LIME approach. This is primarily because LIME relies on generating and classifying a large set of randomly perturbed example images.

## 7    Conclusions

We propose STREAK, the first streaming algorithm for maximizing weakly submodular functions, and prove that it achieves a constant factor approximation assuming a random stream order. This is useful when the set function is not submodular and, additionally, takes a long time to evaluate or has a very large ground set. Conversely, we show that under a worst case stream order no algorithm with memory sublinear in the ground set size has a constant factor approximation. We formulate interpretability of black-box neural networks as set function maximization, and show that STREAK provides interpretable explanations faster than previous approaches. We also show experimentally that STREAK trades off accuracy and running time in nonlinear sparse regression.

One interesting direction for future work is to tighten the bounds of Theorems 5.1 and 5.5, which are nontrivial but somewhat loose. For example, there is a gap between the theoretical guarantee of the state-of-the-art algorithm for submodular functions and our bound for $\gamma = 1$. However, as our algorithm performs the same computation as that state-of-the-art algorithm when the function is submodular, this gap is solely an analysis issue. Hence, the real theoretical performance of our algorithm is better than what we have been able to prove in Section 5.

## 8    Acknowledgments

This research has been supported by NSF Grants CCF 1344364, 1407278, 1422549, 1618689, ARO YIP W911NF-14-1-0258, ISF Grant 1357/16, Google Faculty Research Award, and DARPA Young Faculty Award (D16AP00046).

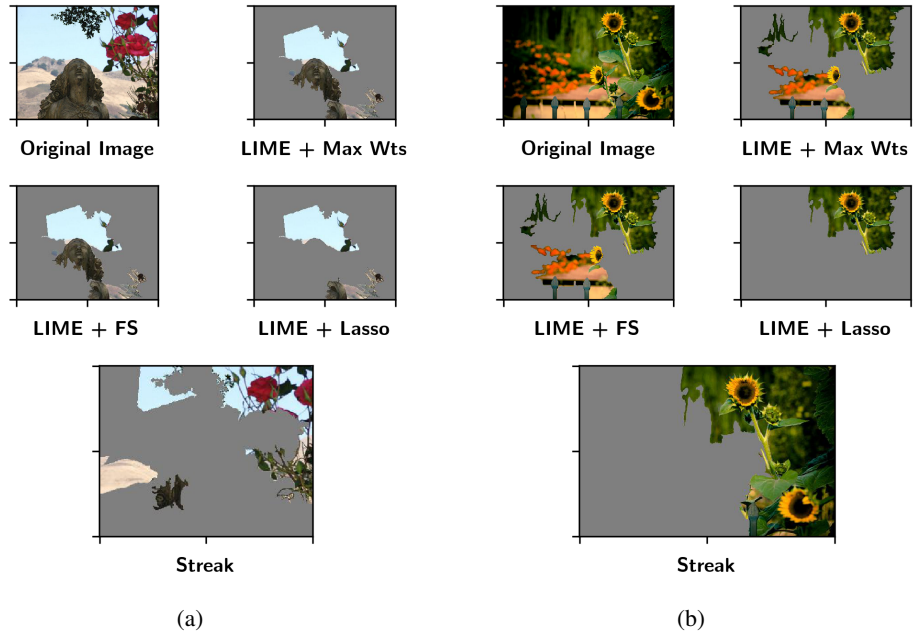

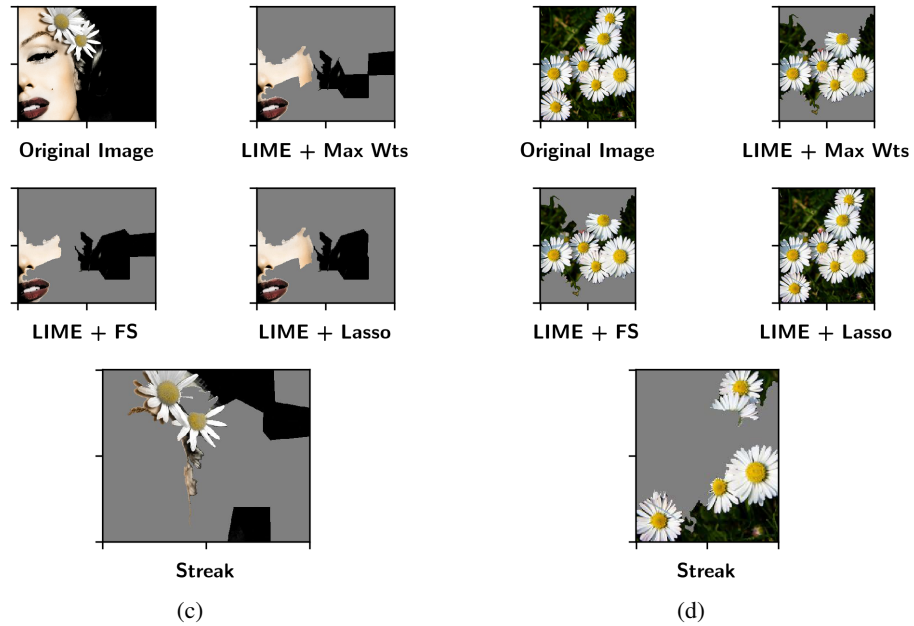

Figure 3: Comparison of interpretability algorithms for the Inception V3 deep neural network. We have used transfer learning to extract features from Inception and train a flower classifier. In these four input images the flower types were correctly classified (from (a) to (d): rose, sunflower, daisy, and daisy). We ask the question of interpretability: *why* did this model classify this image as rose. We are using our framework (and the recent prior work LIME [Ribeiro et al., 2016]) to see which parts of the image the neural network is looking at for these classification tasks. As can be seen STREAK correctly identifies the flower parts of the images while some LIME variations do not. More importantly, STREAK is creating subsampled images on-the-fly, and hence, runs approximately 10 times faster. Since interpretability tasks perform multiple calls to the black-box model, the running times can be quite significant.

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
