[Supplementary Material · streaming-nips-supplemental.pdf]

# A   Supplementary Material

## A.1   Proof of Lemma 4.1

The nonnegativity and monotonicity of $f_k$ follow immediately from the fact that $u(S)$ and $v(S)$ have these properties. Thus, it remains to prove that $f_k$ is 0.5-weakly submodular for $|\mathcal{N}_k|$, *i.e.*, that for every pair of arbitrary sets $S, L \subseteq \mathcal{N}_k$ it holds that

$$\sum_{w \in S \setminus L} f_k(w \mid L) \geq 0.5 \cdot f_k(S \mid L) \ .$$

There are two cases to consider. The first case is that $f_k(L) = 2 \cdot u(L) + 1$. In this case $S \setminus L$ must contain at least $\lceil f_k(S \mid L)/2 \rceil$ elements of $\{u_i\}_{i=1}^k$. Additionally, the marginal contribution to $L$ of every element of $\{u_i\}_{i=1}^k$ which does not belong to $L$ is at least 1. Thus, we get

$$\sum_{w \in S \setminus L} f_k(w \mid L) \geq \sum_{w \in (S \setminus L) \cap \{u_i\}_{i=1}^k} f_k(w \mid L) \geq |(S \setminus L) \cap \{u_i\}_{i=1}^k|$$
$$\geq \lceil f_k(S \mid L)/2 \rceil \geq 0.5 \cdot f_k(S \mid L) \ .$$

The second case is that $f_k(L) = 2 \cdot v(L)$. In this case $S \setminus L$ must contain at least $\lceil f_k(S \mid L)/2 \rceil$ elements of $\{v_i\}_{i=1}^k$, and in addition, the marginal contribution to $L$ of every element of $\{v_i\}_{i=1}^k$ which does not belong to $L$ is at least 1. Thus, we get in this case again

$$\sum_{w \in S \setminus L} f_k(w \mid L) \geq \sum_{w \in (S \setminus L) \cap \{v_i\}_{i=1}^k} f_k(w \mid L) \geq |(S \setminus L) \cap \{v_i\}_{i=1}^k|$$
$$\geq \lceil f_k(S \mid L)/2 \rceil \geq 0.5 \cdot f_k(S \mid L) \ . \qquad \square$$

## A.2   Proof of Theorem 4.2

Consider an arbitrary (randomized) streaming algorithm ALG aiming to maximize $f_k(S)$ subject to the cardinality constraint $|S| \leq 2k$. Since $ALG$ uses $o(N)$ memory, we can guarantee, by choosing a large enough $d$, that ALG uses no more than $(c/4) \cdot N$ memory. In order to show that ALG performs poorly, consider the case that it gets first the elements of $\{u_i\}_{i=1}^k$ and the dummy elements (in some order to be determined later), and only then it gets the elements of $\{v_i\}_{i=1}^k$. The next lemma shows that some order of the elements of $\{u_i\}_{i=1}^k$ and the dummy elements is bad for ALG.

**Lemma A.1.** *There is an order for the elements of $\{u_i\}_{i=1}^k$ and the dummy elements which guarantees that in expectation* ALG *returns at most $(c/2) \cdot k$ elements of $\{u_i\}_{i=1}^k$.*

*Proof.* Let $W$ be the set of the elements of $\{u_i\}_{i=1}^k$ and the dummy elements. Observe that the value of $f_k$ for every subset of $W$ is 0. Thus, ALG has no way to differentiate between the elements of $W$ until it views the first element of $\{v_i\}_{i=1}^k$, which implies that the probability of every element $w \in W$ to remain in ALG's memory until the moment that the first element of $\{v_i\}_{i=1}^k$ arrives is determined only by $w$'s arrival position. Hence, by choosing an appropriate arrival order one can guarantee that the sum of the probabilities of the elements of $\{u_i\}_{i=1}^k$ to be at the memory of ALG at this point is at most

$$\frac{kM}{|W|} \leq \frac{k(c/4) \cdot N}{k + d} = \frac{k(c/4) \cdot (2k + d)}{k + d} \leq \frac{kc}{2} \ ,$$

where $M$ is the amount of memory ALG uses. $\qquad \square$

The expected value of the solution produced by ALG for the stream order provided by Lemma A.1 is at most $ck + 1$. Hence, its approximation ratio for $k > 1/c$ is at most

$$\frac{ck + 1}{2k} = \frac{c}{2} + \frac{1}{2k} < c \ . \qquad \square$$

## A.3 Proof of Observation 5.3

Algorithm 1 adds an element $u$ to the set $S$ only when the marginal contribution of $u$ with respect to $S$ is at least $\tau/k$. Thus, it is always true that

$$f(S) \geq \frac{\tau \cdot |S|}{k} \ . \qquad \qquad \square$$

## A.4 Proof of Proposition 5.4

We begin by proving several intermediate lemmas. Recall that $\gamma \triangleq \gamma_k$, and notice that by the monotonicity of $f$ we may assume that $OPT$ is of size $k$. For every $0 \leq i \leq |OPT| = k$, let $OPT_i$ be the random set consisting of the last $i$ elements of $OPT$ according to the input order. Note that $OPT_i$ is simply a uniformly random subset of $OPT$ of size $i$. Thus, we can lower bound its expected value as follows.

**Lemma A.2.** *For every $0 \leq i \leq k$, $\mathbb{E}[f(OPT_i)] \geq [1 - (1 - \gamma/k)^i] \cdot f(OPT)$.*

*Proof.* We prove the lemma by induction on $i$. For $i = 0$ the lemma follows from the nonnegativity of $f$ since

$$f(OPT_0) \geq 0 = [1 - (1 - \gamma/k)^0] \cdot f(OPT) \ .$$

Assume now that the lemma holds for some $0 \leq i - 1 < k$, and let us prove it holds also for $i$. Since $OPT_{i-1}$ is a uniformly random subset of $OPT$ of size $i - 1$, and $OPT_i$ is a uniformly random subset of $OPT$ of size $i$, we can think of $OPT_i$ as obtained from $OPT_{i-1}$ by adding to this set a uniformly random element of $OPT \setminus OPT_{i-1}$. Taking this point of view, we get, for every set $T \subseteq OPT$ of size $i - 1$,

$$
\begin{aligned}
\mathbb{E}[f(OPT_i) \mid OPT_{i-1} = T] &= f(T) + \frac{\sum_{u \in OPT \setminus T} f(u \mid T)}{|OPT \setminus T|} \\
&\geq f(T) + \frac{1}{k} \cdot \sum_{u \in OPT \setminus T} f(u \mid T) \\
&\geq f(T) + \frac{\gamma}{k} \cdot f(OPT \setminus T \mid T) \\
&= \left(1 - \frac{\gamma}{k}\right) \cdot f(T) + \frac{\gamma}{k} \cdot f(OPT) \ ,
\end{aligned}
$$

where the last inequality holds by the $\gamma$-weak submodularity of $f$. Taking expectation over the set $OPT_{i-1}$, the last inequality becomes

$$
\begin{aligned}
\mathbb{E}[f(OPT_i)] &\geq \left(1 - \frac{\gamma}{k}\right) \mathbb{E}[f(OPT_{i-1})] + \frac{\gamma}{k} \cdot f(OPT) \\
&\geq \left(1 - \frac{\gamma}{k}\right) \cdot \left[1 - \left(1 - \frac{\gamma}{k}\right)^{i-1}\right] \cdot f(OPT) + \frac{\gamma}{k} \cdot f(OPT) \\
&= \left[1 - \left(1 - \frac{\gamma}{k}\right)^i\right] \cdot f(OPT) \ ,
\end{aligned}
$$

where the second inequality follows from the induction hypothesis. $\qquad \square$

Let us now denote by $o_1, o_2, \ldots, o_k$ the $k$ elements of $OPT$ in the order in which they arrive, and, for every $1 \leq i \leq k$, let $S_i$ be the set $S$ of Algorithm 1 immediately before the algorithm receives $o_i$. Additionally, let $A_i$ be an event fixing the arrival time of $o_i$, the set of elements arriving before $o_i$ and the order in which they arrive. Note that conditioned on $A_i$, the sets $S_i$ and $OPT_{k-i+1}$ are both deterministic.

**Lemma A.3.** *For every $1 \leq i \leq k$ and event $A_i$, $\mathbb{E}[f(o_i \mid S_i) \mid A_i] \geq (\gamma/k) \cdot [f(OPT_{k-i+1}) - f(S_i)]$, where $OPT_{k-i+1}$ and $S_i$ represent the deterministic values these sets take given $A_i$.*

*Proof.* By the monotonicity and $\gamma$-weak submodularity of $f$, we get

$$\sum_{u \in OPT_{k-i+1}} f(u \mid S_i) \geq \gamma \cdot f(OPT_{k-i+1} \mid S_i)$$

$$= \gamma \cdot [f(OPT_{k-i+1} \cup S_i) - f(S_i)]$$
$$\geq \gamma \cdot [f(OPT_{k-i+1}) - f(S_i)] \ .$$

Since $o_i$ is a uniformly random element of $OPT_{k-i+1}$, even conditioned on $A_i$, the last inequality implies

$$\mathbb{E}[f(o_i \mid S_i) \mid A_i] = \frac{\sum_{u \in OPT_{k-i+1}} f(u \mid S_i)}{k - i + 1}$$

$$\geq \frac{\sum_{u \in OPT_{k-i+1}} f(u \mid S_i)}{k}$$

$$\geq \frac{\gamma \cdot [f(OPT_{k-i+1}) - f(S_i)]}{k} \ . \qquad \square$$

Let $\Delta_i$ be the increase in the value of $S$ in the iteration of Algorithm 1 in which it gets $o_i$.

**Lemma A.4.** *Fix $1 \leq i \leq k$ and event $A_i$, and let $OPT_{k-i+1}$ and $S_i$ represent the deterministic values these sets take given $A_i$. If $f(S_i) < \tau$, then $\mathbb{E}[\Delta_i \mid A_i] \geq [\gamma \cdot f(OPT_{k-i+1}) - 2\tau]/k$.*

*Proof.* Notice that by Observation 5.3 the fact that $f(S_i) < \tau$ implies that $S_i$ contains less than $k$ elements. Thus, conditioned on $A_i$, Algorithm 1 adds $o_i$ to $S$ whenever $f(o_i \mid S_i) \geq \tau/k$, which means that

$$\Delta_i = \begin{cases} f(o_i \mid S_i) & \text{if } f(o_i \mid S_i) \geq \tau/k \ , \\ 0 & \text{otherwise} \ . \end{cases}$$

One implication of the last equality is

$$\mathbb{E}[\Delta_i \mid A_i] \geq \mathbb{E}[f(o_i \mid S_i) \mid A_i] - \tau/k \ ,$$

which intuitively means that the contribution to $\mathbb{E}[f(o_i \mid S_i) \mid A_i]$ of values of $f(o_i \mid S_i)$ which are too small to make the algorithm add $o_i$ to $S$ is at most $\tau/k$. The lemma now follows by observing that Lemma A.3 and the fact that $f(S_i) < \tau$ guarantee

$$\mathbb{E}[f(o_i \mid S_i) \mid A_i] \geq (\gamma/k) \cdot [f(OPT_{k-i+1}) - f(S_i)]$$
$$> (\gamma/k) \cdot [f(OPT_{k-i+1}) - \tau]$$
$$\geq [\gamma \cdot f(OPT_{k-i+1}) - \tau]/k \ . \qquad \square$$

We are now ready to put everything together and get a lower bound on $\mathbb{E}[\Delta_i]$.

**Lemma A.5.** *For every $1 \leq i \leq k$,*

$$\mathbb{E}[\Delta_i] \geq \frac{\gamma \cdot [\Pr[\mathcal{E}] - (1 - \gamma/k)^{k-i+1}] \cdot f(OPT) - 2\tau}{k} \ .$$

*Proof.* Let $\mathcal{E}_i$ be the event that $f(S_i) < \tau$. Clearly $\mathcal{E}_i$ is the disjoint union of the events $A_i$ which imply $f(S_i) < \tau$, and thus, by Lemma A.4,

$$\mathbb{E}[\Delta_i \mid \mathcal{E}_i] \geq [\gamma \cdot \mathbb{E}[f(OPT_{k-i+1}) \mid \mathcal{E}_i] - 2\tau]/k \ .$$

Note that $\Delta_i$ is always nonnegative due to the monotonicity of $f$. Thus,

$$\mathbb{E}[\Delta_i] = \Pr[\mathcal{E}_i] \cdot \mathbb{E}[\Delta_i \mid \mathcal{E}_i] + \Pr[\bar{\mathcal{E}}_i] \cdot \mathbb{E}[\Delta_i \mid \bar{\mathcal{E}}_i] \geq \Pr[\mathcal{E}_i] \cdot \mathbb{E}[\Delta_i \mid \mathcal{E}_i]$$
$$\geq [\gamma \cdot \Pr[\mathcal{E}_i] \cdot \mathbb{E}[f(OPT_{k-i+1}) \mid \mathcal{E}_i] - 2\tau]/k \ .$$

It now remains to lower bound the expression $\Pr[\mathcal{E}_i] \cdot \mathbb{E}[f(OPT_{k-i+1}) \mid \mathcal{E}_i]$ on the rightmost hand side of the last inequality.

$$\Pr[\mathcal{E}_i] \cdot \mathbb{E}[f(OPT_{k-i+1}) \mid \mathcal{E}_i] = \mathbb{E}[f(OPT_{k-i+1})] - \Pr[\bar{\mathcal{E}}_i] \cdot \mathbb{E}[f(OPT_{k-i+1}) \mid \bar{\mathcal{E}}_i]$$

$$\geq [1 - (1 - \gamma/k)^{k-i+1} - (1 - \Pr[\mathcal{E}_i])] \cdot f(OPT)$$

$$\geq [\Pr[\mathcal{E}] - (1 - \gamma/k)^{k-i+1}] \cdot f(OPT)$$

where the first inequality follows from Lemma A.2 and the monotonicity of $f$, and the second inequality holds since $\mathcal{E}$ implies $\mathcal{E}_i$ which means that $\Pr[\mathcal{E}_i] \geq \Pr[\mathcal{E}]$ for every $1 \leq i \leq k$. $\qquad \square$

Proposition 5.4 follows quite easily from the last lemma.

*Proof of Proposition 5.4.* Lemma A.5 implies, for every $1 \leq i \leq \lceil k/2 \rceil$,

$$
\begin{aligned}
\mathbb{E}[\Delta_i] &\geq \frac{\gamma}{k} f(OPT)[\Pr[\mathcal{E}] - (1 - \gamma/k)^{k - \lceil k/2 \rceil + 1}] - \frac{2\tau}{k} \\
&\geq \frac{\gamma}{k} f(OPT)[\Pr[\mathcal{E}] - (1 - \gamma/k)^{k/2}] - \frac{2\tau}{k} \\
&\geq \left( \gamma \cdot [\Pr[\mathcal{E}] - e^{-\gamma/2}] \cdot f(OPT) - 2\tau \right) / k .
\end{aligned}
$$

The definition of $\Delta_i$ and the monotonicity of $f$ imply together

$$
\mathbb{E}[f(S)] \geq \sum_{i=1}^{b} \mathbb{E}[\Delta_i]
$$

for every integer $1 \leq b \leq k$. In particular, for $b = \lceil k/2 \rceil$, we get

$$
\begin{aligned}
\mathbb{E}[f(S)] &\geq \frac{b}{k} \cdot \left( \gamma \cdot [\Pr[\mathcal{E}] - e^{-\gamma/2}] \cdot f(OPT) - 2\tau \right) \\
&\geq \frac{1}{2} \cdot \left( \gamma \cdot [\Pr[\mathcal{E}] - e^{-\gamma/2}] \cdot f(OPT) - 2\tau \right) . \qquad \square
\end{aligned}
$$

## A.5 Proof of Theorem 5.1

In this section we combine the previous results to prove Theorem 5.1. Recall that Observation 5.2 and Proposition 5.4 give two lower bounds on $\mathbb{E}[f(S)]$ that depend on $\Pr[\mathcal{E}]$. The following lemmata use these lower bounds to derive another lower bound on this quantity which is independent of $\Pr[\mathcal{E}]$. For ease of the reading, we use in this section the shorthand $\gamma' = e^{-\gamma/2}$.

**Lemma A.6.** $\mathbb{E}[f(S)] \geq \frac{\tau}{2a}(3 - \gamma' - 2\sqrt{2 - \gamma'}) = \frac{\tau}{a} \cdot \frac{3 - e^{-\gamma/2} - 2\sqrt{2 - e^{-\gamma/2}}}{2}$ *whenever* $\Pr[\mathcal{E}] \geq 2 - \sqrt{2 - \gamma'}$.

*Proof.* By the lower bound given by Proposition 5.4,

$$
\begin{aligned}
\mathbb{E}[f(S)] &\geq \frac{1}{2} \cdot \{\gamma \cdot [\Pr[\mathcal{E}] - \gamma'] \cdot f(OPT) - 2\tau\} \\
&\geq \frac{1}{2} \cdot \left\{ \gamma \cdot \left[ 2 - \sqrt{2 - \gamma'} - \gamma' \right] \cdot f(OPT) - 2\tau \right\} \\
&= \frac{1}{2} \cdot \left\{ \gamma \cdot \left[ 2 - \sqrt{2 - \gamma'} - \gamma' \right] \cdot f(OPT) - (\sqrt{2 - \gamma'} - 1) \cdot \frac{\tau}{a} \right\} \\
&\geq \frac{\tau}{2a} \cdot \left\{ 2 - \sqrt{2 - \gamma'} - \gamma' - \sqrt{2 - \gamma'} + 1 \right\} \\
&= \frac{\tau}{a} \cdot \frac{3 - \gamma' - 2\sqrt{2 - \gamma'}}{2} ,
\end{aligned}
$$

where the first equality holds since $a = (\sqrt{2 - \gamma'} - 1)/2$, and the last inequality holds since $a\gamma \cdot f(OPT) \geq \tau$. $\qquad \square$

**Lemma A.7.** $\mathbb{E}[f(S)] \geq \frac{\tau}{2a}(3 - \gamma' - 2\sqrt{2 - \gamma'}) = \frac{\tau}{a} \cdot \frac{3 - e^{-\gamma/2} - 2\sqrt{2 - e^{-\gamma/2}}}{2}$ *whenever* $\Pr[\mathcal{E}] \leq 2 - \sqrt{2 - \gamma'}$.

*Proof.* By the lower bound given by Observation 5.2,

$$
\begin{aligned}
\mathbb{E}[f(S)] &\geq (1 - \Pr[\mathcal{E}]) \cdot \tau \geq \left( 1 - 2 + \sqrt{2 - \gamma'} \right) \cdot \tau \\
&= \left( \sqrt{2 - \gamma'} - 1 \right) \cdot \frac{\sqrt{2 - \gamma'} - 1}{2} \cdot \frac{\tau}{a} = \frac{3 - \gamma' - 2\sqrt{2 - \gamma'}}{2} \cdot \frac{\tau}{a} . \qquad \square
\end{aligned}
$$

Combining Lemmata A.6 and A.7 we get the theorem. $\qquad \square$

## A.6 Proof of Theorem 5.5

There are two cases to consider. If $\gamma < 4/3 \cdot k^{-1}$, then we use the following simple observation.

**Observation A.8.** *The final value of the variable $m$ is $f^{\max} \triangleq \max\{f(u) \mid u \in \mathcal{N}\} \geq \frac{\gamma}{k} \cdot f(OPT)$.*

*Proof.* The way $m$ is updated by Algorithm 2 guarantees that its final value is $f^{\max}$. To see why the other part of the observation is also true, note that the $\gamma$-weak submodularity of $f$ implies

$$
\begin{aligned}
f^{\max} &\geq \max\{f(u) \mid u \in OPT\} = f(\varnothing) + \max\{f(u \mid \varnothing) \mid u \in OPT\} \\
&\geq f(\varnothing) + \frac{1}{k} \sum_{u \in OPT} f(u \mid \varnothing) \geq f(\varnothing) + \frac{\gamma}{k} f(OPT \mid \varnothing) \geq \frac{\gamma}{k} \cdot f(OPT) \ . \qquad \square
\end{aligned}
$$

By Observation A.8, the value of the solution produced by STREAK is at least

$$
\begin{aligned}
f(u_m) = m &\geq \frac{\gamma}{k} \cdot f(OPT) \geq \frac{3\gamma^2}{4} \cdot f(OPT) \\
&\geq (1-\varepsilon)\gamma \cdot \frac{3(\gamma/2)}{2} \cdot f(OPT) \\
&\geq (1-\varepsilon)\gamma \cdot \frac{3 - 3e^{-\gamma/2}}{2} \cdot f(OPT) \\
&\geq (1-\varepsilon)\gamma \cdot \frac{3 - e^{-\gamma/2} - 2\sqrt{2 - e^{-\gamma/2}}}{2} \cdot f(OPT) \ ,
\end{aligned}
$$

where the second to last inequality holds since $1 - \gamma/2 \leq e^{-\gamma/2}$, and the last inequality holds since $e^{-\gamma} + e^{-\gamma/2} \leq 2$.

It remains to consider the case $\gamma \geq 4/3 \cdot k^{-1}$, which has a somewhat more involved proof. Observe that the approximation ratio of STREAK is 1 whenever $f(OPT) = 0$ because the value of any set, including the output set of the algorithm, is nonnegative. Thus, we can safely assume in the rest of the analysis of the approximation ratio of Algorithm 2 that $f(OPT) > 0$.

Let $\tau^*$ be the maximal value in the set $\{(1-\varepsilon)^i \mid i \in \mathbb{Z}\}$ which is not larger than $a\gamma \cdot f(OPT)$. Note that $\tau^*$ exists by our assumption that $f(OPT) > 0$. Moreover, we also have $(1-\varepsilon) \cdot a\gamma \cdot f(OPT) < \tau^* \leq a\gamma \cdot f(OPT)$. The following lemma gives an interesting property of $\tau^*$. To understand the lemma, it is important to note that the set of values for $\tau$ in the instances of Algorithm 1 appearing in the final collection $I$ is deterministic because the final value of $m$ is always $f^{\max}$.

**Lemma A.9.** *If there is an instance of Algorithm 1 with $\tau = \tau^*$ in $I$ when STREAK terminates, then in expectation STREAK has an approximation ratio of at least*

$$
(1-\varepsilon)\gamma \cdot \frac{3 - e^{-\gamma/2} - 2\sqrt{2 - e^{-\gamma/2}}}{2} \ .
$$

*Proof.* Consider a value of $\tau$ for which there is an instance of Algorithm 1 in $I$ when Algorithm 2 terminates, and consider the moment that Algorithm 2 created this instance. Since the instance was not created earlier, we get that $m$ was smaller than $\tau/k$ before this point. In other words, the marginal contribution of every element that appeared before this point to the empty set was less than $\tau/k$. Thus, even if the instance had been created earlier it would not have taken any previous elements.

An important corollary of the above observation is that the output of every instance of Algorithm 1 that appears in $I$ when STREAK terminates is equal to the output it would have had if it had been executed on the entire input stream from its beginning (rather than just from the point in which it was created). Since we assume that there is an instance of Algorithm 1 with $\tau = \tau^*$ in the final collection $I$, we get by Theorem 5.1 that the expected value of the output of this instance is at least

$$
\frac{\tau^*}{a} \cdot \frac{3 - e^{-\gamma/2} - 2\sqrt{2 - e^{-\gamma/2}}}{2} > (1-\varepsilon)\gamma \cdot f(OPT) \cdot \frac{3 - e^{-\gamma/2} - 2\sqrt{2 - e^{-\gamma/2}}}{2} \ .
$$

The lemma now follows since the output of STREAK is always at least as good as the output of each one of the instances of Algorithm 1 in its collection $I$. $\qquad \square$

We complement the last lemma with the next one.

**Lemma A.10.** *If $\gamma \geq 4/3 \cdot k^{-1}$, then there is an instance of Algorithm 1 with $\tau = \tau^*$ in $I$ when* STREAK *terminates.*

*Proof.* We begin by bounding the final value of $m$. By Observation A.8 this final value is $f^{\max} \geq \frac{\gamma}{k} \cdot f(OPT)$. On the other hand, $f(u) \leq f(OPT)$ for every element $u \in \mathcal{N}$ since $\{u\}$ is a possible candidate to be $OPT$, which implies $f^{\max} \leq f(OPT)$. Thus, the final collection $I$ contains an instance of Algorithm 1 for every value of $\tau$ within the set

$$\left\{ (1-\varepsilon)^i \mid i \in \mathbb{Z} \quad \text{and} \quad (1-\varepsilon) \cdot f^{\max}/(9k^2) \leq (1-\varepsilon)^i \leq f^{\max} \cdot k \right\}$$
$$\supseteq \left\{ (1-\varepsilon)^i \mid i \in \mathbb{Z} \quad \text{and} \quad (1-\varepsilon) \cdot f(OPT)/(9k^2) \leq (1-\varepsilon)^i \leq \gamma \cdot f(OPT) \right\} .$$

To see that $\tau^*$ belongs to the last set, we need to verify that it obeys the two inequalities defining this set. On the one hand, $a = (\sqrt{2 - e^{-\gamma/2}} - 1)/2 < 1$ implies

$$\tau^* \leq a\gamma \cdot f(OPT) \leq \gamma \cdot f(OPT) \ .$$

On the other hand, $\gamma \geq 4/3 \cdot k^{-1}$ and $1 - e^{-\gamma/2} \geq \gamma/2 - \gamma^2/8$ imply

$$\begin{aligned}
\tau^* &> (1-\varepsilon) \cdot a\gamma \cdot f(OPT) = (1-\varepsilon) \cdot (\sqrt{2 - e^{-\gamma/2}} - 1) \cdot \gamma \cdot f(OPT)/2 \\
&\geq (1-\varepsilon) \cdot (\sqrt{1 + \gamma/2 - \gamma^2/8} - 1) \cdot \gamma \cdot f(OPT)/2 \\
&\geq (1-\varepsilon) \cdot (\sqrt{1 + \gamma/4 + \gamma^2/64} - 1) \cdot \gamma \cdot f(OPT)/2 \\
&= (1-\varepsilon) \cdot (\sqrt{(1 + \gamma/8)^2} - 1) \cdot \gamma \cdot f(OPT)/2 \geq (1-\varepsilon) \cdot \gamma^2 \cdot f(OPT)/16 \\
&\geq (1-\varepsilon) \cdot f(OPT)/(9k^2) \ . \qquad \qquad \square
\end{aligned}$$

Combining Lemmata A.9 and A.10 we get the desired guarantee on the approximation ratio of STREAK. $\square$

### A.7 Proof of Theorem 5.6

Observe that STREAK keeps only one element ($u_m$) in addition to the elements maintained by the instances of Algorithm 1 in $I$. Moreover, Algorithm 1 keeps at any given time at most $\mathcal{O}(k)$ elements since the set $S$ it maintains can never contain more than $k$ elements. Thus, it is enough to show that the collection $I$ contains at every given time at most $\mathcal{O}(\varepsilon^{-1} \log k)$ instances of Algorithm 1. If $m = 0$ then this is trivial since $I = \varnothing$. Thus, it is enough to consider the case $m > 0$. Note that in this case

$$\begin{aligned}
|I| &\leq 1 - \log_{1-\varepsilon} \frac{mk}{(1-\varepsilon)m/(9k^2)} = 2 - \frac{\ln(9k^3)}{\ln(1-\varepsilon)} \\
&= 2 - \frac{\ln 9 + 3\ln k}{\ln(1-\varepsilon)} = 2 - \frac{\mathcal{O}(\ln k)}{\ln(1-\varepsilon)} \ .
\end{aligned}$$

We now need to upper bound $\ln(1 - \varepsilon)$. Recall that $1 - \varepsilon \leq e^{-\varepsilon}$. Thus, $\ln(1-\varepsilon) \leq -\varepsilon$. Plugging this into the previous inequality gives

$$|I| \leq 2 - \frac{\mathcal{O}(\ln k)}{-\varepsilon} = 2 + \mathcal{O}(\varepsilon^{-1} \ln k) = \mathcal{O}(\varepsilon^{-1} \ln k) \ . \qquad \qquad \square$$

## A.8 Additional Experiments

Figure 4: In addition to the experiment in Section 6.2, we also replaced LIME's default feature selection algorithms with STREAK and then fit the same sparse regression on the selected superpixels. This method is captioned "LIME + Streak." Since LIME fits a series of nested regression models, the corresponding set function is guaranteed to be monotone, but is not necessarily submodular. We see that results look qualitatively similar and are in some instances better than the default methods. However, the running time of this approach is similar to the other LIME algorithms.

Original Image     LIME + Max Wts     Original Image     LIME + Max Wts

Streak     LIME + Streak     Streak     LIME + Streak

(a)           (b)

Figure 5: Here we used the same setup described in Figure 4, but compared explanations for predicting 2 different classes for the same base image: 5(a) the highest likelihood label (sunflower) and 5(b) the second-highest likelihood label (rose). All algorithms perform similarly for the sunflower label, but our algorithms identify the most rose-like parts of the image.