[Reviews · NeurIPS 2017]

Reviewer 1



This paper proposes a new approach STREAK for maximizing weakly submodular functions. The idea is to collect several outputs of the Threshold Greedy algorithm, where the selection is based on a given threshold. The theoretical results of the Threshold Greedy algorithm and STREAK are verified sequentially. STREAK is also used to provide interpretable explanations for neural-networks and the empirical studies are given. This is an interesting work. The streaming algorithm is novel and the analyses are elaborate. The problem constructed to prove the ineffectiveness of randomized streaming algorithms is ingenious. The experiments also show the superiority of STREAK. However, how to choose the proper \epsilon in STEAK? Figure 2(a) shows that by varying \epsilon, the algorithm can achieve a gradual tradeoff between speed and performance, but the result is trivial because with the increase of \epsilon, the time and space complexity will both increase and lead to better performance. I have checked all the proofs, and believe that most of them are correct. However, there are also some places which are not clear. 1. In the proof of Lemma A.1, the choice of appropriate arrival order is unclear and how to derive the inequality at the end of the proof? 2. In the proof of Lemma A.9, what is the meaning of the first paragraph? It seems to have nothing to do with the proof in the second paragraph. 3. In the proof of Theorem 5.6, why m=0 implies I=\ emptyset?

Reviewer 2



This paper studies the problem of maximizing functions that are approximately submodular in a streaming setting. The authors develop the STREAK algorithm, which achieves a constant factor approximation for weakly submodular functions with O(k log k) memory when the elements arrive in a random order in the stream. This result is complemented with an impossibility result for streams with adversarial ordering: no algorithm can obtain a constant factor approximation for 1/2-weakly submodular functions. Weakly submodular functions arise in nonlinear sparse regression applications where the function is not submodular. Experimental evaluations show how this algorithm can be used to interpret neural networks in a streaming setting, obtaining explanations of similar quality compared to previous work, but much faster. The problem of extending known theoretical guarantees for submodular optimization problems to more general classes of functions is an important direction and this paper achieves a nice constant factor approximation for maximizing the well-motivated class of weakly submodular functions in a streaming setting, which has been heavily studied for submodular functions. This constant approximation is especially strengthened by the strong impossibility result in the worst case order. The application of this problem to interpreting neural networks is also very interesting and the paper is overall very well written.

Reviewer 3



This paper considers the streaming weakly submodular maximization problem and propose a novel algorithm, STREAK which obtains constant factor guarantees. 1. The authors propose that no randomized streaming algorithm that uses o(N) memory to solve max_{|S| < 2k} f_k(S) has a constant approximation ratio for a worst-case stream order. Does this apply to offline algorithms? If you have o(N) memories, it is possible to store all elements first. 2. As for the bounds in theorem 5.1 and 5.5, I would suggest the author make some plots to show them. Also, when gamma goes to 1 (i.e. the function is submodular), the bounds are 0.18 and 0.016, which is relative low compared to the existing algorithms on submodular online max. Is it possible for STREAK to meeting the performance of THRESHOLD GREEDY on boundary cases (gamma = 1)? 3. What is the theoretical running time complexity of STREAK? 4. For most submodular function in the paper, It should be mentioned that f(\emptyset) \geq 0; otherwise, f(OPT) might be less than 0.